# Safety of Immunization for Children with Immune Thrombocytopenia

**DOI:** 10.3390/vaccines12010066

**Published:** 2024-01-09

**Authors:** Xiangshi Wang, Tianxing Feng, Chuning Wang, Jingjing Li, Yanling Ge, Xiaowen Zhai, Hongsheng Wang, Mei Zeng

**Affiliations:** 1Department of Infectious Disease, Children’s Hospital of Fudan University, National Children’s Medical Center, Shanghai 201102, China; shi_16811@163.com (X.W.);; 2Department of Pediatrics, Shanghai Clinical Research and Trial Center, Shanghai 201203, China; tianxing@fudan.edu.cn; 3Department of Hematology, Children’s Hospital of Fudan University, National Children’s Medical Center, Shanghai 201102, China; xwzhai@fudan.edu.cn

**Keywords:** immune thrombocytopenia, vaccine-associated thrombocytopenia, vaccine hesitancy, catch-up immunization, children

## Abstract

Vaccine hesitancy is a common issue for children with immune thrombocytopenia (ITP) in China. The objective of this paper is to assess the immunization statuses of children with ITP, analyze the possible relationship between immunization and thrombocytopenia, and evaluate the safety of immunization after ITP remission. We included 186 children with an ITP history and followed up with them for two years after receiving re-immunization recommendations. The participants had an overall age-appropriate vaccine coverage of 57.9%. Vaccine-associated thrombocytopenia occurred in 99 (53.2%, 95% CI = 46.06–60.26) children ranging from 0 to 34 days following immunization, with 14 vaccines involved. One hundred and fifty-four (82.3%, 95% CI = 76.72–87.54) children were advised to restart immunization, whereas 32 (17.2%, 95% CI = 12.46–23.28) were advised to postpone partial or full vaccination. Following the follow-up, 150 (80.6%, 95% CI = 74.37–85.68) children completed the catch-up immunization, whereas 27 (14.5%, 95% CI = 10.17–20.30) partially completed it. Four patients with thrombocytopenia relapsed following the re-immunization. Incomplete catch-up immunization was related to the factors of chronic thrombocytopenia, vaccine-associated thrombocytopenia, and the relapse of ITP following re-immunization. ITP may occur after immunization with vaccines other than measles-containing vaccines. Re-immunization in children with ITP generally does not result in a relapse, regardless of whether the previous thrombocytopenia was vaccine-associated.

## 1. Introduction

Immune thrombocytopenia (ITP) can be an isolated primary condition, or it may occur secondary to certain diseases, drugs, or vaccines [1]. It has been hypothesized that vaccine-associated thrombocytopenia is mediated by antigen-induced specific autoimmunity [2]. Theoretically, there is a risk of redeveloping thrombocytopenia when the body is exposed to the same antigen. Parents and healthcare providers may hesitate to assent to administering the incident vaccine and even all subsequent immunizations due to concerns about a relapse of thrombocytopenia after re-immunization. The available data show that vaccine-associated thrombocytopenia is very rare. The surveillance data on adverse events following immunization have revealed that there were 773 cases of thrombocytopenia in China during 2010–2015, with a reported incidence ranging from 0.20 to 0.35 cases per million doses [3]. Similarly, 1440 cases of thrombocytopenia were reported by the US Vaccine Adverse Event Reporting System during 1990–2008 [4]. Vaccine-associated thrombocytopenia may occur 1 to 30 days after almost all vaccinations [5], with a relatively high incidence in measles-containing vaccines of 3.3 per million children per year, while the incidence for diphtheria and tetanus toxoids and pertussis and haemophilus influenzae type B and inactive polio vaccines is 1.2, and that for 13-valent pneumococcal polysaccharide conjugate vaccines is 1.2, all within the Chinese population [6]. Although vaccines can induce thrombocytopenia, the definite mechanism has not been fully established so far. Only the first dose of the measles–mumps–rubella vaccine has been demonstrated to have a clear causal relationship with the occurrence of ITP with some biological plausibility, given that the measles virus itself can cause ITP [7]. The other notable aspect is a possible association between ITP and COVID-19 vaccines, specifically the ChadOx1-S (AstraZeneca) vaccine, noting that this was in an adult population and not seen with mRNA vaccines [8]. The incidence of ITP peaks in early childhood, which often overlaps with the period when the National Immunization Program (NIP) is widely implemented. The data from European countries have been used to estimate that the incidence of acute ITP in children is 1.9–6.4 per million children per year [9]. This figure is at least 10 times higher than that of reported post-immunization ITP. Post-immunization ITP is usually a coincident adverse event following immunization (AEFI). Moreover, parents also worry if immunization will trigger a relapse of thrombocytopenia. 

Vaccine hesitancy is a common issue for children with medical conditions in China. Childhood vaccine-associated thrombocytopenia usually has a favorable prognosis [10]. The incidence of ITP is significantly lower than that observed during the natural course of vaccine-preventable diseases [11]. ITP is not listed as a contraindication for any vaccine’s administration in the international as well as Chinese practice guidelines for immunization [12,13,14,15]. The Advisory Committee on Immunization Practices suggested that there were no contraindications for MMR administration to children with a history of ITP who are unimmunized with the MMR and other vaccines [16]. Despite the above evidence, immunization delay or discontinuation after ITP onset is a common issue in China. The uncertainty of relapse risk dampens parental confidence in vaccination. Thus, we carried out this prospective study to assess the immunization statuses of children with immune thrombocytopenia, analyze the possible relationship between immunization and thrombocytopenia, and actively monitor the safety of re-immunization after immune thrombocytopenia remission.

## 2. Materials and Methods

### 2.1. Study Subjects

This study was conducted in the Immunization Advisory Clinic at a National Children’s Medical Center in Shanghai, constituting the first specialized Immunization Advisory Clinic in China. We enrolled children < 18 years old who had a definite diagnosis of thrombocytopenia and were confirmed to be afflicted with ITP utilizing the Brighton Collaboration case definition [17]. The children were brought to the clinic by their parents or were referred by primary care providers or pediatric hematologists for immunization consultation from 2017 to 2019. We collected detailed information about thrombocytopenia and immunization history. We reassessed the possible causes of thrombocytopenia and provided recommendations on immunization. To observe relapses of thrombocytopenia following re-immunization, we actively followed up with the counseled children for 2 years. Ethical approval was granted by the hospital’s Institutional Review Board. Written consent was waived because the data were obtained during routine medical service and follow-ups. Data pooling and analyses were anonymously conducted, without involving any privacy concerns.

### 2.2. Data Collection

Clinical data on medical conditions and immunization history were obtained using a standardized spreadsheet during the on-site parent interview and a review of medical record documents conducted by two physicians providing immunization counselling. The causes of thrombocytopenia and discharge diagnoses were reviewed by two immunization specialists and two hematologists. The patients’ information was collected, including sex, age, date of symptom onset, platelet counts during the disease course, exposure to vaccines within 42 days before thrombocytopenia-related symptom onset, illness within 6 weeks before ITP diagnosis, drug treatment after diagnosis, previous health status, and up-to-date vaccination records. ITP is classified as newly diagnosed (0–3 months), persistent (>3–12 months), or chronic (>12 months) [12]. Severe ITP refers to Grade 4 bleeding determined using an updated bleeding scale for pediatric patients with ITP; this grade is defined as mucosal bleeding leading to a decrease in hemoglobin levels > 2 g/dL or suspected or confirmed internal hemorrhage. Remission of ITP is defined as a blood platelet count ≥ 100,000 per μL and no new mucosal bleeding or visceral hemorrhages for at least 3 months. Trained specialists evaluated the causal relationship between ITP and vaccines according to the World Health Organization manual for the surveillance of adverse events following immunization [18]. Vaccine-associated thrombocytopenia was considered to be present if there was a time correlation between thrombocytopenia occurrence and vaccination within 42 days of vaccine administration; simultaneously, a preceding infectious illness or the concomitant administration of drugs were excluded.

### 2.3. Confirmation of Immunization Records

Immunization history, including vaccine name, vaccine type, and date of administration, was verified during on-site consultation through checking the carry-on immunization handbook. Children were excluded from this study if their parents could not provide the immunization record handbook for verification.

Age-appropriate vaccine coverage included the 23 doses of 14 NIP vaccines mandatorily required by the age of six. The data were recorded using a standardized spreadsheet. The following were the recommended immunization schedules for children aged 0–6 in China in 2016 [19]: 1 dose of the Bacillus Calmette–Guérin vaccine (BCG) at birth; 3 doses of the hepatitis B vaccine (HepB) at birth and at 1 and 6 months of age; 4 doses of the inactive poliomyelitis vaccine (IPV) or bivalent oral poliomyelitis attenuated live vaccine (bOPV) at 2, 3, 4, and 48 months of age; 4 doses of the diphtheria, tetanus, and acellular pertussis combined vaccine (DTaP) at 3, 4, 5, and 18 months of age; 1 dose of the diphtheria and tetanus combined vaccine (DT) at the age of 6 years; 2 does of a measles-containing vaccine (MCV), including the measles attenuated live vaccine (MV), the measles and rubella combined attenuated live vaccine (MR), and the measles, mumps, and rubella combined attenuated live vaccine (MMR), at 8 and 18 months of age; 2 doses of the Group A meningococcal polysaccharide vaccine (MPV-A) at 6 and 9 months of age; 2 doses of the Group A and C meningococcal polysaccharide vaccine (MPV-AC) at 3 and 6 years of age; 2 doses of the Japanese encephalitis attenuated live vaccine (JEV-L) at 8 and 24 months; and 2 doses of the hepatitis A inactivated vaccine (HepA-I) at 18 and 24 months. (Table 1) When a self-paid vaccine was given to replace the free NIP vaccine, the free vaccine was recorded as an immunization.

### 2.4. Immunization Recommendation and Follow-Up

To harmonize management across the team in the Immunization Advisory Clinic, a practical protocol for immunization recommendation for children with ITP was developed after a review of the literature and current guidelines [12,13,14,15]. Children with ITP are recommended to restart or resume immunization when this disease has been in remission for at least 3 months. If it has been recommended that live vaccines should be withheld, the parents will be informed of the vaccine type and interval according to the blood product and immunosuppressive therapy used. It is recommended that live vaccines (bOPV, MCV, JEV-L, and varicella vaccines) should be withheld in the following scenarios [14]: (1) at least 30 days after discontinuation or tapering of corticosteroid use when using corticosteroids at a dose of ≥2 mg/kg per day of prednisone or its equivalent, or ≥20 mg per day if the weight is ≥10 kg for 14 consecutive days or more; (2) at least 14 days after discontinuation or tapering of corticosteroid use when having administered corticosteroids at a dose of ≥2 mg/kg per day of prednisone or its equivalent, or ≥20 mg per day if the weight is ≥10 kg for less than 14 consecutive days or at a dose of <2 mg/kg per day; and (3) 3 months after having administered cyclosporine at ≥2.5 mg/kg per day or mycophenolate mofetil at ≥30 mg/kg per day. It is recommended that live vaccines of the MCV and varicella vaccines (VarVs) should be deferred at least 6 months after transfusion of packed red blood cells or whole blood, 7 months after plasma or platelet cell transfusion, 8 months after therapy consisting of intramuscular or intravenous immunoglobin (IVIG) administration at a total dose of 400 mg/kg, or 10 and 11 months after total doses of 1000 and 2000 mg/kg, respectively, for the previous treatment. The children were vaccinated at their local community immunization clinics. We conducted active telephone follow-ups and passive surveillance based on parents’ self-reports to monitor relapses of ITP following immunization. A relapse of ITP was defined as a platelet count < 100,000 per μL. All subjects were followed up with for two years to record updated immunization statuses via either telephone or WeChat. Children with confirmed secondary ITP other than vaccine-associated thrombocytopenia during the follow-up period were also excluded.

### 2.5. Statistical Analyses

For continuous variables, descriptive statistics were reported as means (standard deviations) or medians (interquartile ranges), and for nominal variables, they were reported as quantities of cases. Shapiro–Wilk and Q–Q plots were used to test normality. Student’s *t*-test was used to compare regularly distributed data, while the Wilcoxon Rank Sum test was used to analyze non-normally distributed data. Categorical variables were reported as percentages and compared using either Pearson’s χ^2^ test or Fisher’s exact test. To determine the statistical significance between the vaccine-unassociated and vaccine-associated thrombocytopenia subgroups, as well as between the receiving-all-catch-up-vaccines subgroup and the not-receiving-all catch-up-vaccines subgroup, Pearson’s χ^2^ test or Fisher’s exact test were used. A *p*-value < 0.05 was considered statistically significant. STATA 16.1 (StataCorp, College Station, TX, USA) was used for the statistical analysis.

## 3. Results

A total of 196 children with thrombocytopenia were screened; three cases were excluded due to hereditary thrombocytopenia, one was excluded for lupus-associated thrombocytopenia, and six were excluded due to not providing complete immunization records (Figure 1). Table 2 shows the characteristics of the 186 included children with ITP examined in this study. One hundred and twenty-seven (68.3%) children were boys, and the median age at the first visit was 21 (IQR: 12, 36) months old. The initial episode of thrombocytopenia occurred at the ages of 0–5 months (n = 100, 53.8%), 6–11 months (n = 44, 23.7%), 12–23 months (n = 27, 14.5%), and ≥24 months (n = 15, 8.1%). These children received observation alone (n = 20), IVIG alone (n = 105), steroid alone (n = 9), and IVIG and steroid (n = 52) treatments. Of all the cases, 158 (84.9%) were acute, 18 (9.7%) were persistent, and 10 (5.4%) were chronic. Three (1.6%) cases were defined as serious cases. At their first visit to the clinic, 164 (88.2%) of the patients had been in remission and 22 (11.8%) were still being treated. The male-to-female ratio, age at first visit, age at the onset of disease, the interval from disease onset to first visit, treatment after diagnosis, and disease type exhibited no statistical differences between the vaccine-unassociated and vaccine-associated thrombocytopenia subgroups (Table 2).

Of the 186 children who had ITP, 99 (53.2%, 95% CI = 46.06–60.26) developed thrombocytopenia within 42 days following vaccination, involving 14 NIP vaccines and self-paid vaccines such as the VarV, enterovirus 71 inactivated vaccine (EV71), oral rotavirus vaccine (ORV), and 13-valent pneumococcal polysaccharide conjugate vaccine (PPCV13) (Table 3). The vaccines related to thrombocytopenia were as follows: the IPV for 32 children, MCV for 18 children, DTaP and HepB for 16 children, and JEV-L for 13 children.

Table 4 shows the impact of thrombocytopenia events on vaccination. Among the 186 children, 137 (73.7%) discontinued all immunization due to health providers’ refusal or parental concern, 20 (10.8%) stopped receiving live vaccines, 9 (4.8%) stopped receiving vaccines suspected to be related to thrombocytopenia, 11 (5.9%) received irregular immunization, and only 9 (4.8%) continued age-appropriate immunization. Vaccination discontinuation was more common in the vaccine-unassociated thrombocytopenia subgroup compared to the vaccine-associated thrombocytopenia subgroup (81.6% vs. 66.7%, *p* < 0.05), whereas live vaccines were more commonly discontinued in the vaccine-associated thrombocytopenia group (17.2% vs. 3.4%, *p* < 0.05).

After counselling with specialists, 154 (82.3%, 95% CI = 76.72–87.54) children were recommended to restart immunization (the restarting immunization group), while 32 (17.2%, 95% CI = 12.46–23.28) were recommended to defer reception of all vaccines (deferring immunization group) due to unstable disease status (n = 8), and others were recommended to defer the reception of the MCV due to the recent use of IVIG (n = 23) and to defer reception of all live vaccines due to the recent administration of packed red blood cells (n = 1) (Figure 1). After two-year follow-ups, 9 children did not receive any vaccines, 150 (80.6%, 95% CI = 74.37–85.68) received all age-appropriate vaccines (the full-catch-up immunization group), and 27 (14.5%, 95% CI = 10.17–20.30) received partial vaccines (the partial-catch-up immunization group). Four children in the partial-catch-up vaccination group experienced a relapse of ITP after restarting immunization and thereafter discontinued subsequent immunization. Of these four relapse cases, one child who had a newly diagnosed case of ITP with a temporally related varicella vaccine had a relapse after hand, foot, and mouth disease; one child with previous persistent ITP had a relapse after HepB vaccination; one child with previous chronic ITP had a relapse after DTaP and OPV administration; and one child with previous chronic ITP had a relapse after vaccination with the MPSV-AC.

Compared to the partial-catch-up immunization group, the full-catch-up immunization group was younger (18.5 vs. 26.5 months old, *p* = 0.003), less frequently received IVIG and steroid combination therapy (22.7% vs. 50.0%, *p* = 0.001), and had fewer chronic ITP cases (1.3% vs. 22.2%, *p* < 0.001), vaccine-associated thrombocytopenia cases (48.7% vs. 72.2%, *p* = 0.011), and episodes of ITP relapse (0% vs. 5.6%, *p* = 0.037) (Table 5).

## 4. Discussion

To the best of our knowledge, this study included the largest number of ITP case series in relation to vaccine safety. Our findings showed that ITP is a big safety concern regarding immunization among parents and healthcare providers, whether related to vaccines or not. Rejection of any type of vaccine or exclusive live vaccine administration are common for children with a history of ITP. Our follow-up findings demonstrated that re-immunization after ITP remission was safe for children and generally did not cause a disease relapse.

Thrombocytopenia is an uncommon adverse event following vaccine administration. Vaccine-associated thrombocytopenia is induced by autoantibodies after binding to a specific platelet glycoprotein (GP), including GP Ib/IX, GP V, and GP IIb/IIIa [20]. Okazaki reported that anti-measles and anti-rubella virus IgG antibodies in platelets were isolated in an ITP-afflicted child after receiving an MMR [21]. This illness is more prevalent in young children since their idiotypic networks are still developing, increasing the likelihood of cross-reactive autoantibody production following immunization [11]. In this study, we found that almost all vaccines were possibly linked with the occurrence of ITP. Most of the cases of vaccine-associated thrombocytopenia occurred after vaccination with the IPV, followed by the DTaP, HepB, and MR. However, the vaccines most reported to be associated with ITP were the VarV in the United States and the MMR in Canada; additionally, the DTaP and HepA were also commonly reported [4,5]. Based on the national surveillance data on AEsFI in China, adverse events relating to ITP have mostly been reported after the MMR, IPV, and HepB vaccinations [6]. However, the VarV and trivalent inactivated influenza vaccine (TIV) were less reported for causing ITP in China than in the United States. This difference may have been because the VarV and TIV are self-paid vaccines, and their coverage is low in Chinese children. We also observed a few cases of ITP secondary to vaccinations with the JEV-L and MPV-A. The use of the Japanese encephalitis inactivated vaccine (JEV-I) and meningococcal polysaccharide combined vaccine (MPCV) may have minimized the incidence of ITP, as only two cases of ITP were reported after vaccination with the MPCV and no cases of ITP were reported after vaccination with the JEV-I according to the 2018–2020 national surveillance data on AEsFI [6,22,23]. Because the link between ITP and vaccines other than the MMR was based on a small number of vaccine-exposed case reports, the causal relationship requires more investigation for determination [24]. Platelet-reactive autoantibodies may also be induced by viral or environmental factors in the event of vaccine-unassociated thrombocytopenia. Antibody-coated platelets are prematurely destroyed in the spleen, liver, or both through interaction with Fcγ receptors [25]. Autoantibodies can also induce the complement-mediated or desialylation-induced destruction of platelets [26,27] and impede megakaryocyte function [28]. When antiplatelet antibodies are not detected, T cell abnormalities such as the skewing of T helper cells toward type 1 helper T and type 17 helper T phenotypes [29] and a reduction in the number and function of regulatory T cells [30] have been described, which could drive the autoimmune process.

Overall, it is safe to start re-immunization for children with ITP after the disease has entered remission and is stable. After two years of follow-up, none of the children in the vaccine-associated thrombocytopenia subgroup experienced an ITP relapse following re-immunization, while three children with persistent or chronic ITP in the vaccine-unassociated thrombocytopenia subgroup experienced an ITP relapse. Although a few case reports documented recurrent ITP following the same or different vaccinations [31,32], our findings demonstrated that revaccination for children with prior ITP and the administration of the incident vaccine and other vaccines to children with previous vaccine-associated thrombocytopenia did not result in the recurrence or progression of thrombocytopenia. Moreover, the prognosis was usually favorable for children who suffered an episode of vaccine-associated thrombocytopenia. Our study also demonstrated that the children in the vaccine-associated thrombocytopenia subgroup were less likely to develop persistent and chronic ITP than those in the vaccine-unassociated thrombocytopenia subgroup. Hsieh reported that all 12 patients with vaccine-associated thrombocytopenia investigated in their study recovered within 6 days, and no recurrence was observed even after receiving the same vaccination later on [33]. Persistent low levels of platelet antibodies in the bodies of children with persistent or chronic ITP may render them prone to reactivation by vaccines [34].

Professional immunization consulting did, in fact, help to address the vaccination safety issue. All children with a previous history of ITP were informed that there were no contraindications for immunization. Due to recent blood product use or immunosuppressive therapy, several children were recommended to postpone receiving all or a portion of injectable live virus vaccines. During the two-year follow-up, most of the counseled children completed a catch-up immunization regimen. We proposed that the appropriate timing of restarting immunization was three months after the platelet count returns to the normal range. Moreover, we did not advise parents to routinely follow up the platelet count after each vaccination except for the first reimmunization to avoid unnecessary healthcare visits and testing. Our findings will assist hematologists and general practitioners in making proper immunization recommendations to their patients with ITP. Parents will acquire a great deal of confidence and reassurance from an on-site immunization consultation. For children who have developed thrombocytopenia following immunization, parents’ concerns about immunization safety are particularly serious. Thus, active follow-ups and full communication with parents will improve vaccination adherence in line with specialists’ recommendations.

Although this was a single-center study, the counseled children came from all over the country; therefore, the participants in this study were representative. Our findings indeed reflected a worldwide phenomenon of vaccination concerns among parents as well as practitioners concerning the link between immunization and ITP. This study contributes more trustworthy evidence of vaccine safety and unveils the role of specialists in addressing vaccine concerns. One limitation was the feasibility of seeing a vaccine specialist for all ITP patients and providing follow-up care for these patients. The study also lacked a qualitative component, making it difficult to remark on the unique concerns of healthcare providers and parents. This could be identified as an issue that will require further research.

## 5. Conclusions

The administration of vaccines to children with a previous history of ITP, whether due to vaccine-associated thrombocytopenia or not, was safe and well-tolerated, and vaccination could be resumed after remission. The immunization compliance was satisfactory, and a relapse of ITP following vaccination was very rare. Consultation with specialized immunization professionals should be encouraged to address vaccine safety concerns.

## Figures and Tables

**Figure 1 vaccines-12-00066-f001:**
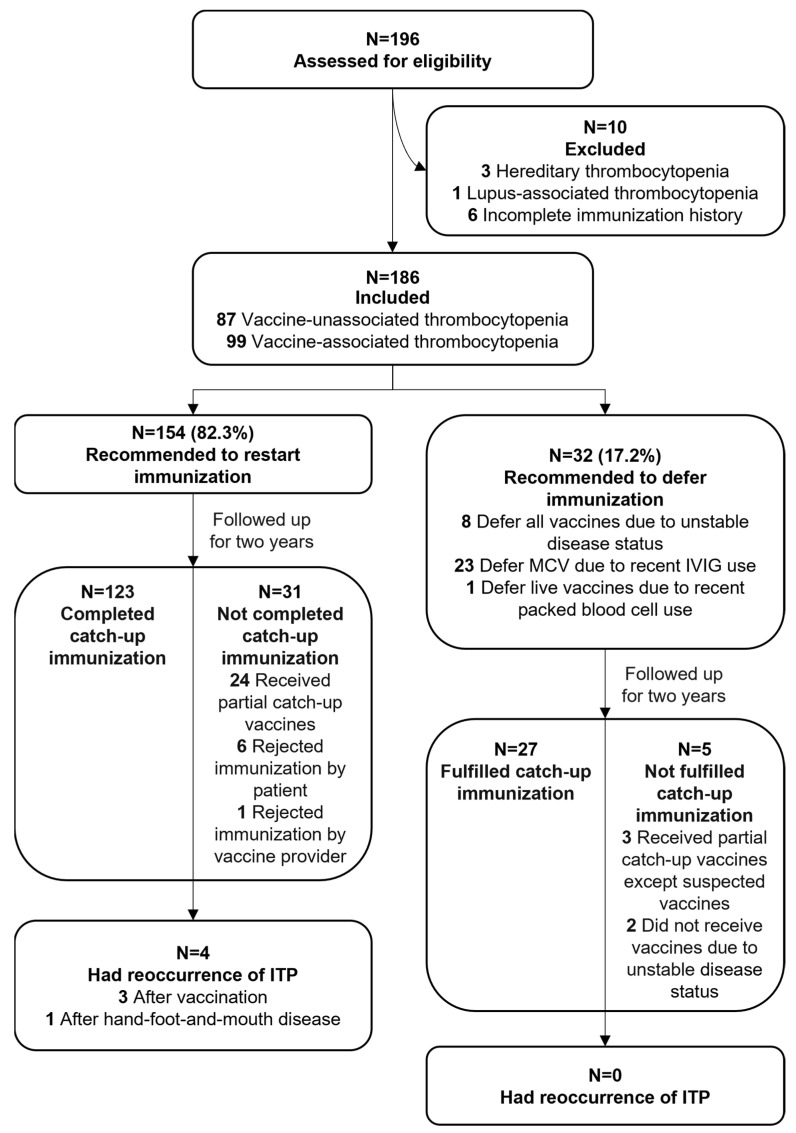
Flowchart of data on immune thrombocytopenic purpura participants throughout study. MCVs: measles-containing vaccines; IVIG: intravenous immunoglobin; ITP: immune thrombocytopenia.

**Table 1 vaccines-12-00066-t001:** Recommended immunization schedule for children aged 6 years or younger in China.

	Birth	1 mos	2 mos	3 mos	4 mos	5 mos	6 mos	8 mos	9 mos	18 mos	2 yrs	3 yrs	4 yrs	6 yrs
Bacillus Calmette–Guérin vaccine	1st dose													
Hepatitis B vaccine	1st dose	2nd dose					3rd dose							
Inactive poliomyelitis vaccine			1st dose											
Bivalent oral poliomyelitis attenuated live vaccine				1st dose	2nd dose								3rd dose	
Diphtheria, tetanus, and acellular pertussis combined vaccine				1st dose	2nd dose	3rd dose				4th dose				
Diphtheria and tetanus combined vaccine														1st dose
Measles and rubella combined attenuated live vaccine								1st dose						
Measles, mumps, and rubella combined attenuated live vaccine										1st dose				
Group A meningococcal polysaccharide vaccine							1st dose		2nd dose					
Group A and C meningococcal polysaccharide vaccine												1st dose		2nd dose
Japanese encephalitis attenuated live vaccine								1st dose			2nd dose			
Hepatitis A inactivated vaccine										1st dose	2nd dose			

**Table 2 vaccines-12-00066-t002:** Clinical characteristics of children with ITP in this study (N = 186).

	Total n = 186	Vaccine-Unassociated Thrombocytopenia n = 87	Vaccine-Associated Thrombocytopenia n = 99	*p* Value
Male	127 (68.3)	58 (66.7)	69 (70.0)	0.658
Age at first visit, month	21 (12, 36)	23 (12, 45)	19 (12, 32)	0.288
Age at disease onset, month	4 (2, 10)	6 (2, 14)	3 (2, 8)	0.205
0–5 mo	100 (53.8)	40 (46.0)	60 (60.6)	
6–11 mo	44 (23.6)	17 (19.5)	27 (27.3)	
12–23 mo	27 (14.5)	17 (19.5)	10 (10.1)	
≥24 mo	15 (8.1)	13 (14.9)	2 (2.0)	
Interval from disease onset to first visit, month	12 (7, 23.8)	12 (7, 24)	12 (6, 26)	0.910
Platelet count at diagnosis, per μL	13.5 (7, 25.5)	17.5 (10, 40)	10 (5.5, 20)	0.006
Treatment after diagnosis				0.407
Observation	20 (10.8)	12 (13.8)	8 (8.1)	
IVIG alone	105 (56.4)	44 (50.6)	61 (61.6)	
Steroid alone	9 (4.8)	5 (5.7)	4 (4.0)	
IVIG + steroid	52 (28.0)	26 (29.9)	26 (26.3)	
Diagnosis type ^a^				0.124
Newly diagnosed	158 (84.9)	69 (79.3)	89 (89.9)	
Persistent	18 (9.7)	12 (13.8)	6 (6.1)	
Chronic	10 (5.4)	6 (6.9)	4 (4.0)	
Serious ITP case ^b^	3 (1.6)	3 (3.4)	0 (0)	0.100
Disease in remission ^c^				0.002
Yes	164 (88.2)	70 (80.5)	94 (94.9)	
No	22 (11.8)	17 (19.5)	5 (5.1)	

IVIG: intravenous immunoglobulin; ITP: immune thrombocytopenic purpura. Data are presented as means (SD) for normally distributed data and as medians (IQR) for skewed data. ^a^ Immune thrombocytopenic purpura was classified as newly diagnosed if the disease presented within the last three months, persistent if the disease was present for 3 to 12 months, and chronic if the disease lasted more than 12 months. ^b^ A serious ITP case was defined as mucosal hemorrhage resulting in a >2 g/dL decrease in hemoglobin levels or suspected or confirmed visceral hemorrhage. Variables were compared using Fisher’s exact test. ^c^ Remission of ITP was defined as a blood platelet count ≥ 100,000 per μL for at least 3 months and no new instances of mucosal bleeding or visceral hemorrhages.

**Table 3 vaccines-12-00066-t003:** Days from vaccination to disease onset in the subgroup of vaccine-associated immune thrombocytopenia cases (N = 99).

Vaccine Type	Number of Involving Dose	Days from Vaccination to Disease Onset, Median (Range)
IPV	32	7 (0–30)
DTaP	16	7 (0–20)
HepB	16	13 (1–28)
MR	16	19.5 (4–34)
JEV-L	13	19.5 (2–34)
MPV-A	8	24 (1–34)
bOPV	6	7 (0–20)
BCG	4	14.5 (5–21)
EV71	4	6.5 (1–14)
HepA-I	4	18.5 (3–30)
VarV	4	17 (1–25)
PPCV13	3	7 (4–21)
ORV	2	10.5 (0–21)
MMR	2	10 (3–17)

A total of 99 cases of vaccine-associated immune thrombocytopenia were reported, involving 14 vaccines and 130 doses. If a patient was inoculated with more than one vaccine within 6 weeks before the disease’s onset, the vaccine administered closest to the disease’s onset was assumed to be the trigger. If more than one vaccine was administered at one time, each vaccine was counted as an individual trigger.

**Table 4 vaccines-12-00066-t004:** Vaccine hesitancy patterns of children with ITP after initial diagnosis in this study.

	Vaccine-Unassociated Thrombocytopenia n = 87	Vaccine-Associated Thrombocytopenia n = 99	*p* Value
No. of children who stopped receiving all vaccines	71 (81.6)	66 (66.7)	0.021
No. of children who only stopped receiving live vaccines ^a^	3 (3.4)	17 (17.2)	0.003
No. of children who stopped receiving incident vaccines ^b^	0 (0)	9 (9.1)	0.004
No. of children who received irregular immunization	8 (9.2)	3 (3.0)	0.117
No. of children who received all types of immunization ^b^	5 (5.7)	4 (4.0)	0.736

The level of significance determined using the Bonferroni Test was established at 0.01. ^a^ Live vaccines refer to the bivalent oral poliomyelitis attenuated live vaccine, measles-containing vaccines, and the Japanese encephalitis attenuated live vaccine. ^b^ Variables were compared using Fisher’s exact test.

**Table 5 vaccines-12-00066-t005:** Comparison of characteristics between two groups completing all age-appropriate catch-up immunizations and partial age-appropriate catch-up immunizations during the two-year follow-up (N = 186).

	Completing All Catch-Up Immunizations n = 150	Completing Partial Catch-Up Immunizations n = 36	*p* Value
Male	102 (68.0)	25 (69.4)	0.876
Age at first visit, month	18.5 (10, 32)	26.5 (19.5, 43.5)	0.003
Age at disease onset, month	4 (2, 11)	5 (2, 8)	0.962
Interval from disease onset to first visit, month	11 (6, 18.5)	18.5 (12, 34.8)	<0.001
Platelet count at diagnosis, per μL	15 (7, 32)	10 (6, 18)	0.082
Treatment of IVIG + steroid	34 (22.7)	18 (50.0)	0.001
Chronic ITP case ^a^	2 (1.3)	8 (22.2)	<0.001
Serious ITP case ^b,c^	5 (3.3)	1 (2.8)	1.000
Case of disease status in treatment ^c^	17 (11.3)	5 (13.9)	0.773
Vaccine-associated thrombocytopenia case	73 (48.7)	26 (72.2)	0.011
Relapse of ITP ^c^	0 (0)	2 (5.6)	0.037

IVIG: intravenous immunoglobulin; ITP: immune thrombocytopenic purpura. Data are presented as means (SD) for normally distributed data and as medians (IQR) for skewed data. ^a^ Chronic immune thrombocytopenic purpura is defined as disease lasting more than 12 months. ^b^ A serious ITP case is defined as mucosal hemorrhage resulting in a >2 g/dL decrease in hemoglobin levels or suspected or confirmed visceral hemorrhage. ^c^ Variables were compared using Fisher’s exact test.

## Data Availability

The data presented in this study are available on request from the corresponding author. The data are not publicly available due to privacy restrictions.

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
