# Peer review of "Safety of Immunization for Children with Immune Thrombocytopenia"

_vaccines, 2024, doi:10.3390/vaccines12010066_

Round 1
Reviewer 1 Report
Comments and Suggestions for Authors
Comments for authors:
In this study, Wang et al assessed the immunization status of children with immune thrombocytopenia and analyzed the possible relationship between immunization and thrombocytopenia. They tracked the safety of immunization after immune thrombocytopenia remission in 186 patients with a previous history of immune thrombocytopenia. They observed that immune thrombocytopenia occurred following immunization of various vaccines besides measles-containing vaccine. However, reimmunization in children with immune thrombocytopenia does not generally result in a relapse, regardless of whether previous thrombocytopenia is vaccine-associated. The analysis with a large number of patients is important even though the results have been randomly reported elsewhere.
Major:
The authors analyzed both vaccine-unrelated thrombocytopenia subgroup and vaccine associated thrombocytopenia subgroup. They discussed the development of anti-measles and anti-rubella virus IgG antibodies that bind specifically to platelets, resulting in reduction of platelet counts in the vaccine-associated thrombocytopenia subgroup. This is reasonable to explain the formation of thrombocytopenia in the vaccine-associated thrombocytopenia subgroup. However, no explanation was made for the vaccine-unrelated thrombocytopenia subgroup. How does thrombocytopenia develop in this subgroup? What is the mechanism here and how does it differ from the vaccine associated thrombocytopenia subgroup?
Minor:
The manuscript needs extensive editing. Many errors occur here and there, especially in references. For example:
- Page 4, Line 149: references are strangely cited: ‘….literature and current guideline6, 11, 12, 16-186, 11, 12, 16-18’
- Page 4, Line 154: …in the following scenarios1919
- Page 4, Line 165: …of 2000 mg/kg1919.
- Page 8. Line 274: References: …and GP IIb/IIIa2020
- Page 8. Line 276: References:… after receiving MMR2121
- Many other references are not correctly cited.
- 20mg per dayà A space is required after ‘20’. Please uniform others
- The authors used both ‚μl‘ and ‚µL‘ or ‘ml and mL’, please uniform them
- Page 4, Line 190: …6-11 months à English: required plural: 6-11 months. Please check the whole manuscript.
- Page 4, Line 190: Why do you need ‘respectively’ here? (unnecessary)
- Too many abbreviations were not defined or introduced. This makes the manuscript difficult to read.
- Page 6, Table 3: The first row is very difficult to understand, please improve.
- Page 6, Line 232: no space before ‘b’. Also in Table 4.
Comments on the Quality of English LanguageThe manuscript needs extensive editing.
Reviewer 2 Report
Comments and Suggestions for Authors
The manuscript titled "Safety of Immunization for Children with Immune Thrombocytopenia" analyzes the relationship between vaccination/immunization and thrombocytopenia in children in China. Overall, the manuscript is well written and informative. The results presented do support the claims in the discussion. The authors' findings will be important in educating practitioners on the risks of thrombocytopenia as it relates to vaccination in children. What would be of interest and a consideration to the authors is to continue such evaluations with some of the newer vaccines coming to market such as COVID and mpox. Vaccine hesitancy is still a problem to this day and as new vaccine platforms come to the market, it will be important to provide data on the safety of vaccination.
I have only minor comments for the authors to consider.
Line 83: There appears to be a typo in the citations with references 11,1311, and 13 listed.
Line 101: Please state why 2 years was chosen for followup of the cases.
Table 3: The column headings do not align.
Reviewer 3 Report
Comments and Suggestions for Authors
This is a good study evaluating the incidence of vaccine-induced thrombocytopenia in children with ITP. The text and the tables were clear. However, a few comments are suggested.
Comments on the abstract:
1. The abstract is extremely long. The journal's instructions ask for 200 words. You can also remove the headings from the abstract part (Introduction, Methods, Results, Conclusion) as the abstract should be unstructured per the journal's guidelines.
2. The introduction isn't an introduction. It only includes the study objective. Instead, it should contain a brief introduction into the topic.
3. The number of enrolled children is repeated in both the methods and the results section. It should only be included in the results part of the abstract.
4. "Vaccine associated thrombocytopenia occurred in 99 children": Include the %.
Comments on the main text:
1. 2.3 Confirmation of Immunization Record (lines 130-144): I suggest creating a table summarizing the vacciantion schedule.
2. Line 185: I suggest replacing "was enrolled" with "were screened." Also, I suggest citing Figure 1 at the end of the sentence on line 187.
3. Conclusion (line 338): after "well-tolerated" I suggest adding ", where vaccination can be resumed after remission."
Reviewer 4 Report
Comments and Suggestions for Authors
The manuscript by Wang et al is n interesting paper on the Safety of Immunization for Children with Immune 2 Thrombocytopenia . Some minor changes should be done.
Major Issues
1. The abstract provides percentages. The 95% confidence interval should be included in parentheses. 95% confidence intervals can be easily computed with the freely available Open Epi program. https://www.openepi.com/Proportion/Proportion.htm
2. In material and methods mention how the normality of data was tested (e.g., Shapiro-Wilk test, Q-Q plots), etc.
3. The text mentions using Pearson's 𝜒2 test or Fisher's exact test for comparing subgroups related to thrombocytopenia and vaccination status. When comparing within the group, the authors should consider the problem of multiple comparisons, and they used a correction for multiple testing (like Bonferroni correction) to control for Type I error. It is enough to write in a note at the foot of the table, the level of significance using the Bonferroni Test, was established at eg: p= 0.0125. Another possibility is the contingency tables n x n, to compute adjusted standardized residuals, they will point to what cells are significant.
Minor issues
4. Check the bibliographic references because the first reference in the introduction is number 11. It is likely that for some reason, the reference number seems to be repeated and is the cause of them appearing in number 11 instead of number one. Something similar occurs in the material and methods. For example, the reference to the methodology of the World Health Organization is 1515.
5. In the material and methods, it is preferable to use the variable sex instead of gender since gender refers to sexual orientation, and sex is a biological characteristic. The gender variable should only be used if information on people's sexual orientation is available, and in that case, explain how the information was obtained.
Comments on the Quality of English Language
The first sentence of the abstract sounds strange "Introduction: To assess the immunization status of children with immune thrombocytopenia, analyze the possible relationship between immunization and thrombocytopenia, observe the safety of immunization after immune thrombocytopenia remission. Maybe a verb is missing or should change into "The objective of this paper is o assess the immunization status of children with immune thrombocytopenia, analyze the possible relationship between immunization and thrombocytopenia, observe the safety of immunization after immune thrombocytopenia remission"
Round 2
Reviewer 1 Report
Comments and Suggestions for Authors
- The authors did not significantly improve the manuscript. The writing level is still very low. References are still wrongly cited!
- The authors have added a paragraph to describe the case of 'vaccine unassociated thrombocytopenia': It mentions that platelet-reactive autoantibodies may be triggered by viral or environmental factors in the case of vaccine unassociated thrombocytopenia. Platelets coated with IgG autoantibodies are cleared more quickly in the spleen and liver.'
Is it true? It has been shown for many years that autoantibodies contain an Fc regime that can activate platelets via FcgamaRIIa. This results in platelet activation and aggregations, even thrombotic evens. The autoantibodies can activate also other cells such as neutrophils, monocytes, and bridge endothelial cells. These autoantibodies can cause life-threatening thrombotic events.
The authors wrote: 'Platelets coated with IgG autoantibodies are cleared more quickly in the spleen and liver' This is not true.
Comments on the Quality of English LanguageThe writing level is still very low.
Round 3
Reviewer 1 Report
Comments and Suggestions for Authors The manuscript is now improved but the authors can improve it further. I now recommend publishing it at Vaccines after their careful revision.Comments on the Quality of English Language
/